mental health; adolescent; Sri Lanka; adolescent mental health; Sri Lankan adolescents

**Corresponding author:**
Jane Fisher;
Email: jane.fisher@monash.edu

# Prevalence and determinants of mental health problems experienced by school-going adolescents in Sri Lanka

Chethana Mudunna[1] , Miyuru Chandradasa[2], Thach Duc Tran[1,3] ,
Josefine Antoniades[1,4,5] and Sivunadipathige Sumanasiri[2] and Jane Fisher[1]

[1]Global and Women's Health, Monash School of Public Health and Preventive Medicine, Melbourne, VIC, Australia;
[2]Department of Psychiatry, Faculty of Medicine, University of Kelaniya, Ragama, Sri Lanka; [3]Biostatistics Unit, Faculty of Health, Deakin University, Burwood, VIC, Australia; [4]National Ageing and Research Institute, Parkville, VIC, Australia and [5]School of Humanities & Social Sciences, La Trobe University, Bundoora, VIC, Australia

## Abstract

The mental health of Sri Lankan adolescents is of growing concern, given the decades of internal conflict and socio-political instability in Sri Lanka. This aims were to examine the prevalence and determinants of symptoms of common mental health problems (MHP) experienced by school-going adolescents in Sri Lanka. A cross-sectional survey was conducted among school-going adolescents in grades 10–12/13 from seven schools in Gampaha District, Sri Lanka. Depressive/psychological distress symptoms measured using the PHQ-9 /K10, were analysed using mean scale scoring. Psychosocial determinants were measured using JVQ/PBI/AESI/study-specific questions. Associations between MHPs and psychosocial determinants were examined using multiple linear regression models. 24.11% of 1,045 adolescents who completed the surveys reported clinically significant symptoms of depression, and 60.10% reported psychological distress. Higher age, being female, lesser physical activity, smoking, daily social media use, violent victimisation, not living with both birth parents, having ≥2 siblings, low maternal/paternal education, having an overprotective paternal figure, increased academic stress and rural living were associated with higher MHPs. We identified a high prevalence of MHPs among Sri Lankan adolescents, which was multifactorially determined. Modifiable risk factors addressed through public health policies, research and programmes, as well as less-modifiable risk factors addressed through national-level policy changes, are all essential to addressing mental health burdens in this population.

## Impact statement

This study fills a critical gap in understanding the mental health status of Sri Lankan adolescents, as well as the influence of social and structural determinants on the mental health of Sri Lankan adolescents, a population that faces unique mental health challenges resulting from the remnants of a civil war and decades of socio-political instability. By adopting a framework guided by Bronfenbrenner's socio-ecological model, this study aims to provide actionable insights for educators, caregivers, public health researchers and policymakers seeking to design interventions that can be implemented to address key risk factors and enhance protective factors of adolescent mental health. The findings show that Sri Lankan adolescents reported extremely higher levels of psychological distress (60.1%) in comparison to data from similar populations across high-income and low- and middle-income countries. Modifiable risk factors identified included health risk factors, violent victimisation and factors in the home and school environments of adolescents. Less-modifiable risk factors included structural determinants, such as gender and area of residence. Positive caregiving acted protectively for adolescent mental health. Findings suggest that public health programmes, such as school-based mental health programmes, physical education programmes and caregiver education programmes, can contribute to addressing risk factors and enhancing protective factors. Additionally, large-scale policy measures can work towards addressing structural determinants of mental health. Furthermore, evidence gaps on the association between violent victimisation and adolescent mental health provide a comprehensive roadmap for future research.

The wider impact of this research lies in its potential to inform evidence-based policies, programmes and research that can improve the mental well-being of Sri Lankan adolescents. At a global level, these findings reinforce those of the Lancet Commission on Adolescent Health and Wellbeing, where this study unpacks the strong influence of determinants, that are often beyond individual control on the mental health of adolescents.



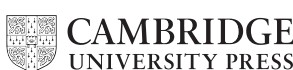

## Introduction

Adolescence, the life phase from ages 10–19 years, is a unique period of human development during which individuals experience rapid formative physical, cognitive, emotional and social growth (Kieling et al., 2011). Globally, it is estimated that one in seven (14%) adolescents experience mental health problems (MHPs), accounting for 13% of the global disease burden in this age group (World Health Organization, 2021). In low- and middle-income countries (LMICs), population-level epidemiological data on the prevalence of MHPs is limited (Fisher et al., 2011) as existing mental health data on children and adolescents in LMICs only cover 2% of the population (United Nations Children's Fund and World Health Organization, 2022). Although data from small-scale studies do indicate that young people in LMICs experience symptoms of depression and anxiety, and that these are multi-factorially determined (Fisher et al., 2011). The mental health of adolescents in LMICs is a pressing matter, particularly to policymakers, given that 90% of the world's adolescent population lives in LMICs (United Nations Children's Fund and World Health Organization, 2022). Additionally, as described in the recent Lancet Commission on Adolescent Health and Wellbeing (Baird et al., 2025), progress in adolescent health in the twenty-first century has been undermined by increased morbidity and mortality in this group resulting from non-communicable diseases, such as MHPs. Therefore, addressing these challenges is vital to prevent their compounding effects. For MHPs, if unaddressed early, these problems can lead to long-term mental health disability in adulthood (United Nations Children's Fund and World Health Organization, 2022).

Sri Lanka is classified by the World Bank as a lower-middle-income country in South Asia with an ethnically diverse and multi-faith population of 22 million people (Shoib et al., 2022; Metreau et al., 2024). One-fifth of the population is adolescents (Rasalingam et al., 2022). For three decades, Sri Lanka was ravaged by armed conflict, which ended in 2009 (Shoib et al., 2022). The end of the conflict was followed by socio-economic instability, regional conflicts and natural disasters (Shoib et al., 2022). Peace in Sri Lanka has been unstable. In 2019, a large-scale terrorist attack reignited trauma felt in post-conflict Sri Lanka (Shoib et al., 2022). Soon after, in 2020, the coronavirus disease 2019 pandemic amplified the mental health burden globally (Shoib et al., 2022). During and immediately after the pandemic, Sri Lanka endured its worst economic crisis since the end of colonial rule, which exacerbated socio-economic and political instability across the country (Shoib et al., 2022). The dual impact of a pandemic and economic crisis resulted in scarcity in food, medicine, fuel, power outages and a significant elevation in the cost of living (Shoib et al., 2022). In an already low-resourced nation, young people in Sri Lanka faced substantial detriments from these events, including most importantly, being deprived of schooling (Shoib et al., 2022). Although little is known about the mental health of Sri Lankan adolescents, it is of growing concern, given the consecutive adverse events the country has experienced (Shoib et al., 2022).

Some evidence suggests that Sri Lankan adolescents have experienced psychosocial stressors as a result of adverse events experienced by the nation as a whole. Post-war era cross-sectional research indicates high levels of depressive and anxiety symptoms at 36 and 26%, respectively, among school-going adolescents (Rodrigo et al., 2010). Moreover, recent studies that analysed data collected by the World Health Organization in 2016 through the Global School-Based Health Survey (GSHS) suggest an overall prevalence of MHPs as high as 40.3% (Rasalingam et al., 2022).

However, post-war data and GSHS data collected in 2016 may not accurately represent the mental health status of Sri Lankan adolescents in the present (Rasalingam et al., 2022). Nevertheless, these findings indicate a high prevalence rate of MHPs among Sri Lankan adolescents in comparison to the global average (Rasalingam et al., 2022).

Mental health is multifaceted and complex. Adolescence adds another layer of complexity. A growing body of evidence highlights that across the life course, the interplay of psychosocial, environmental, cultural, socio-economic, biological and genetic factors contributes to protecting or harming mental health (Kieling et al., 2011; Evans, 2023). During adolescence, which is a particularly sensitive developmental phase, the effect of these factors may be especially impactful, and may directly or indirectly influence the onset and severity of MHPs (Kieling et al., 2011; Lund et al., 2018). For example, adverse social and economic circumstances, such as poverty, can be a significant determinant of an adolescent's mental well-being (Lund et al., 2018).

In Sri Lanka, there is little consolidated evidence about the determinants of adolescent mental health. Data from GSHS examined determinants, such as demographic characteristics, food habits, personal hygiene, behavioural problems, substance use and parental and social engagement (Rasalingam et al., 2022). Findings highlight risk factors towards developing MHPs, such as gender, food insecurity, truancy, second-hand smoking or physical fighting, while protective factors, such as engagement with parents and close friends, adequate nutrition and physical activity, positively correlated with the mental health of Sri Lankan adolescents (Rasalingam et al., 2022). The GSHS captured a small selection of actors. It was concluded that further comprehensive assessment of determinants that span across multiple ecosystems of a Sri Lankan adolescent's immediate and surrounding environment is needed (Rasalingam et al., 2022).

One of the biggest limitations to understanding the mental health needs of Sri Lankan adolescents is the lack of up-to-date data on prevalence and minimal exploration of determinants of MHPs among Sri Lankan adolescents. The aim of this study was to describe the prevalence and determinants of depressive symptoms and psychological distress among school-going adolescents in Sri Lanka.

## Methods

### Study design

A cross-sectional study based on an anonymous survey of adolescent students in Gampaha District, Sri Lanka.

### Study setting

Located in Western Province, Gampaha District is the second most populous district in Sri Lanka (City Population, 2022). A population census in 2021 reported that the district has a population of 2.4 million people (City Population, 2022). Most (90.5%) are Sinhalese, while smaller populations of Sri Lankan Moor (4.2%) and Sri Lankan Tamil (3.9%) groups reside in Gampaha District (City Population, 2022). The adolescent population of this district was recorded as being around 350,000 or 15% of the population in 2012 (City Population, 2022).

Gampaha is divided by the Federal Ministry of Education into four education zones and the zones are divided into educational divisions (Abayasekara and Arunatilake, 2018). Data collection for

this study took place in Gampaha Division, Gampaha zone of Gampaha District in Western Province, Sri Lanka. There are 33 government schools in Gampaha Division (Ministry of Education, 2021).

## Participants

### Inclusion criteria

The inclusion criteria were as follows: Adolescents aged 14–19 years; in grades 10, 11 and 12/13 and in a Sinhala or English medium school in Gampaha District, Sri Lanka.

### Exclusion criteria

Participants were excluded if they were unable to use the online survey platform or printed version of the survey, even with support, due to physical or psychological disability. For rural schools, students absent on the day of paper-based survey administration were excluded.

## Sample size

The required sample size to establish prevalence of MHPs was 844 (Supplementary Material 1).

## Data sources

The cross-sectional survey comprised 6 sections and 33 items. The first section included socio-demographic questions. The second section included two standardised psychometric tools to assess symptoms of depression and psychological distress: The Patient Health Questionnaire (PHQ-9) (Kroenke et al., 2001) and the Kessler Psychological Distress Scale (K10) (Kessler et al., 2002). The third used modules of the Juvenile Victimization Questionnaire (JVQ) (Hamby et al., 2011) to assess experiences of violent victimisation. The fourth section examined young people's experiences of their relationships with their parents using the Parental Bonding Instrument (PBI) (Kendler, 1996). The fifth used the Academic Expectations Stress Inventory (AESI) (Ang and Huan, 2006) to measure academic pressure and school environment. The final section examined protective factors using structured study-specific questions. Details of assessment tools are described in Table 1.

## Recruitment

A multi-stage sampling technique was used to recruit students from seven selected Sinhala and English medium public secondary schools in Gampaha District, Sri Lanka.

First, among secondary schools in Gampaha district, four urban, one semi-urban and two rural schools were selected using a stratified random sampling technique. Second, two classes from each of the year levels 10, 11 and 12/13 were randomly selected by an independent statistician. In schools with both Sinhala and English medium streams of education, a Sinhala and an English medium class were selected from each year level. Finally, all eligible students in these classes were invited to participate in this study.

## Procedure and consent

First, school teachers and principals notified parents of the study. Several Zoom meetings were hosted between school staff, the research team and parents to notify parents about the nature of the study.

Next, consent forms were sent to parents of students under the age of 18 years. All who returned parental signed consent were eligible to participate. Signed assent was obtained from all eligible students, whether over or under 18 years.

The survey was conducted from July to September 2023 via Qualtrics and paper-based methods. In urban and semi-urban schools, the survey was administered online during class time, either in the presence of the primary investigator, a research officer and class teachers, or, in some instances, only in the presence of class teachers. In rural schools, the survey was administered via paper-based methods due to a lack of IT infrastructure. Data collection required multiple visits to schools due to resource limitations and class-time availability. For urban and semi-urban schools, the survey link for each school was provided to teachers of the selected classes to distribute to eligible students via class WhatsApp groups. This enabled eligible students who were unavailable on the day of online survey administration to complete the survey remotely. Detailed information on the consent process and ethical problems is provided in Supplementary Material 1.

## Data management and analysis

Surveys with <50% items completed and surveys where the items for the primary outcome variables were not completed were excluded from analyses. Missing data were managed using pairwise deletion.

The associations between psychosocial determinants and depressive (PHQ9 scores) and psychological distress symptoms (K10 scores) were examined using multiple linear regression models.

A conceptual framework informed by Bronfenbrenner's Socio-ecological Model (Evans, 2023), adapted to the Sri Lankan context, was developed to guide data analysis and interpretation (Supplementary Figure S1). The framework highlights factors in a Sri Lankan adolescent's immediate and surrounding environment that may affect their mental well-being. These factors were measured in the survey and are categorised into six determinant groups for analysis and interpretation: individual factors (age, gender and siblings); health risk factors (physical activity, alcohol consumption, smoking and social media use); violent victimisation (JVQ); family environment (PBI, living environment, maternal and paternal education, maternal and paternal occupation, family happiness and caregiver religiosity) and school environment (AESI) and community (area of residence). The study hypothesised an association between determinants of the outcome variables, namely symptoms of depression and psychological distress. All determinants highlighted in Supplementary Figure S1 were included in the regression models.

## Ethics approval

First, ethics approval was obtained from the University of Kelaniya, Faculty of Medicine Ethics Review Committee (ID: P/124/09/2022). Next, the project was registered with the Monash University Human Research Ethics Committee (ID: 37225). Third, approval to conduct fieldwork was obtained from the Department of Education, Western Province, Sri Lanka (ID: WP/ED/DEV/13/V); the Line Ministry of Education (ID: ED/03/56/02) and Zonal Education Office of Gampaha Zone, Gampaha District, Sri Lanka.

**Table 1.** Assessment tools

| Variables | Tool | Description | Scoring | Interpretation |
|---|---|---|---|---|
| Socio-demographic characteristics | Structured questions developed by the research team | Twenty items asking the following information: age, gender, ethnicity, religion, year level, area of residence, living environment, maternal caregiver, maternal caregiver education level, maternal caregiver occupation, paternal caregiver, paternal caregiver education level, paternal caregiver occupation, parent's religiosity, number of siblings, physical activity, non-physical extra-curricular activity, alcohol consumption, smoking, social media use. | Participants were able to select from multiple-choice answers or write their response when the option they wished to select was unavailable. | Categorised according to answers. |
| **Primary outcome** | | | | |
| Symptoms of depression | Patient Health Questionnaire–9 (PHQ–9) | Nine-item subscale of the larger PHQ is directly based on the diagnostic criteria for major depressive disorder in the Diagnostic and Statistical Manual (DSM-IV). The PHQ–9 has been validated for use among South Asian adolescents aged 10–19 years in India and Bangladesh (Ganguly et al., 2013). It has been translated and formally validated for use among Sri Lankan adults through gold standard structured clinical interviews (Hanwella et al., 2014). Further, the PHQ–9 has also been validated for use among Sri Lankan adolescents aged 15–17 years, although this is unpublished (Madushani and Weeratunga, 2020). Hence, we report symptoms in relation to commonly used cut-off scores internationally (Kroenke et al., 2001). Permission was obtained to use the Sinhala translation of the instrument for this study, while the English version is publicly available. | Four-point rating scale where 0 = *not at all*, 1 = *several days*, 2 = *more than half the days* and 3 = *nearly every day* (Kroenke et al., 2001). Scores can range from 0 to 27. The higher the score, the more severe the symptoms. | Two methods of classification of symptoms. First is severity of symptoms. Non-minimal ≤4, mild 5–9, moderate 10–14, moderately severe 15–19 and severe 20–27 (Kroenke et al., 2001). Second, clinically significant symptoms of depression where scores were ≥10. |
| Symptoms of psychological distress including depressive and anxiety symptoms | The Kessler Psychological Distress Scale (K10) | Ten-item scale. Its psychometric properties have been validated for use among Sri Lankan adults through the "gold standard" method (Wijeratne et al., 2011). As the scale has not been validated for this population in Sri Lanka, we report symptoms according to validation for use in a similar adolescent population in a study conducted in Indonesia (Tran et al., 2019). Permission was obtained to use the Sinhala translation of the instrument for this study, while the English version is publicly available. | Five-point rating scale where 0 = *none of the time*, 1 = *a little of the time*, 2 = *some of the time*, 3 = *most of the time*, 4 = *all of the time* (Tran et al., 2019). Scores can range from 0–40. Higher scores indicate higher levels of psychological distress. | Two methods of classification of symptoms. First is the severity of symptoms. Low or no symptoms 0–5, moderate 6–11, high 12–19 and very high 20–40 (Tran et al., 2019). Second, clinically significant symptoms of psychological distress where scores ≥18. |
| **Secondary outcome** | | | | |
| Violent victimisation | Juvenile Victimization Questionnaire-Revision 2 (JVQ-R2) | The JVQ determines the true burden of victimisation experienced by youth. It has been widely used to measure various forms of victimisation among adolescents aged 10–19 years living in South Asia. For this survey, 4 items of the Child Maltreatment (Module B), 8 items of the Witnessing and Indirect Victimization (Module E) and 6 items of the Exposure to Family Violence (Module G) modules were used (Hamby et al., 2011). This questionnaire has not been validated for use in Sri Lanka, but has been previously used in a Sri Lankan young adult population (Fernando and Karunasekera, 2009). The questionnaire is publicly available and was translated into Sinhala using the standard procedure for the purpose of this study. | Item-level scoring comprised "yes" and "no" options. 1 = *yes*, 0 = *no.* Anyone who answers "yes" to the item is classified as experiencing victimisation (Hamby et al., 2011). | Higher scores indicate higher numbers of participants who have experienced forms of victimisation. |
| Relationship with caregivers | Parental Bonding Instrument (PBI) | The PBI evaluates parental attitudes derived from an individual's childhood experiences | Four-point rating scale where 1 = *very like*, 2 = | Higher scores in each scale indicates either more care/ |

*(Continued)*

**Table 1.** (*Continued*)

| Variables | Tool | Description | Scoring | Interpretation |
|---|---|---|---|---|
| | | with their parents. Respondents reflect on the first 16 years of their life and rate their maternal and paternal figures' behaviours separately. The 16-item short form was used (Kendler, 1996). The scale represents two parenting style factors: care (7 items) and protectiveness/authoritarianism (9 items). The PBI has not been validated for use in Sri Lanka but has been used in a Sri Lankan Tamil adolescent population (Sriskandarajah et al., 2015a). It was translated for use in this present study using the standard procedure. | *somewhat like*, 3 = *a little like* and 4 = *not at all like* (Kendler, 1996). Subscales are scored separately. Scores can range from 7 to 28 for the Care subscale and from 9 to 36 for the Overprotectiveness subscale (Kendler, 1996). | warmth or more overprotection by the respective caregiver. |
| Academic environment | Academic Expectations Stress Inventory (AESI) | AESI measures the specific role of students' expectations and their parents'/teachers' expectations in producing or exacerbating academic stress. It is mostly used among adolescent students aged 11–18 years. This study used the 9-item scale (Ang and Huan, 2006). Higher scores indicate higher academic stress. The AESI has not been previously translated or validated for use in Sri Lanka. It has been translated for use in the present study. | Four-point rating scale was used where 0 = *never*, 1 = *rarely*, 2 = *sometimes* and 3 = *often*. Scores can range from 0 to 27 (Ang and Huan, 2006). | Higher scores indicate higher levels of academic stress. |

## Results

### Socio-demographic characteristics

A total of 1,387 adolescents were eligible to participate, and 1,045 (75.34%) completed the survey. Non-participation was due to parental non-consent, absence from school and student non-consent. Exact numbers of non-participants could not be established as student management was conducted by school teachers. Most participants were female, Sinhalese and Buddhist (Table 2). The average age was 15.9 years (SD 1.13).

### Depressive symptoms and psychological distress

A total of 1,045 adolescents completed the PHQ-9 questionnaire fully, and 1,044 completed the K10 questionnaire fully. The mean total scores of both scales are shown in Table 3. One in four and three in five adolescents had clinically significant symptoms of depression and psychological distress, respectively. PHQ-9 and K10 total scores were further categorised according to the level of severity of depressive and psychological distress symptoms (see Table 3).

### Psychosocial determinants of depression and psychological distress

Results of the multiple linear regression model examining the associations between potential factors and symptoms of depression are shown in Table 4. The $R^2$ value was 0.35. Being female, rarely/not engaging in sporting activities, smoking, daily social media use, experiencing child maltreatment, witnessing/indirect violent victimisation, exposure to family violence, not living with both birth parents, overprotectiveness by paternal figure, increased academic stress and living in a rural area were significantly associated with higher depressive symptoms. Maternal warmth and affection were significantly associated with lower depressive symptoms.

The results of the multiple linear regression model examining the association between potential factors and psychological distress are shown in Table 5. The $R^2$ value was 0.16. Higher age, being female, having two or more siblings, rarely/not engaging in sporting activities, experiencing child maltreatment, witnessing/indirect violent victimisation, not living with both birth parents, maternal and paternal education level of secondary schooling, paternal education level of primary/no schooling, overprotectiveness by paternal figure and increased academic stress were significantly associated with higher psychological distress.

## Discussion

In this study, we identified that one in four Sri Lankan adolescents was experiencing clinically significant symptoms of depression, and more than half had clinically significant symptoms of psychological distress. Risk and protective factors of MHPs included individual, family and wider social characteristics, confirming that these problems are multifactorially determined in this context. To our knowledge, existing research in Sri Lanka had not investigated adolescent mental health using a framework that observes the influence of multiple levels of environmental factors. Findings from this study reinforce the findings from the Lancet Commission on Adolescent Health and Wellbeing by unpacking the influence of structural and social determinants on adolescent mental health.

### Strengths and limitations

Study strengths include rigorous methodology, allowing for reproducibility of the study. An appropriate sample size was obtained to establish prevalence and allow for greater generalisability of the findings to the broader adolescent population in Sri Lanka. Bilingual healthcare professionals translated the survey tool to ensure the accuracy and appropriateness of the language used in the tool for the population assessed. Bilingual members of the research team

**Table 2.** Socio-demographic characteristics of Sri Lankan adolescents (*N* = 1,045)

| Variable | Frequency | Percentage % |
|---|---|---|
| Sex | | |
| Female | 579 | 55.41 |
| Male | 461 | 44.11 |
| Other/prefer not to say | 5 | 0.48 |
| Ethnicity | | |
| Sinhala | 1,037 | 99.23 |
| Muslim/Tamil/Burgher | 8 | 0.77 |
| Religion | | |
| Buddhist | 1,007 | 96.36 |
| Catholic/Christian | 34 | 3.25 |
| Islam/Hindu/no religion | 4 | 0.38 |
| Year level | | |
| 10 | 486 | 46.51 |
| 11 | 319 | 30.53 |
| 12 | 229 | 21.91 |
| 13 | 11 | 1.05 |
| Area of residence | | |
| Urban | 194 | 18.56 |
| Semi-urban | 559 | 53.49 |
| Rural | 292 | 27.94 |
| Living with | | |
| Both of your birth parents | 943 | 90.24 |
| Only one of your birth parents | 89 | 8.52 |
| None of your birth parents | 13 | 1.24 |
| Main female carer | | |
| Birth mother | 1,001 | 95.79 |
| Other family member (grandmother, stepmother, adoptive mother, older sister, other family member)/other non-family member | 44 | 4.21 |
| Maternal figure education | | |
| University degree or higher | 162 | 15.5 |
| Diploma or certificate | 140 | 13.4 |
| Secondary school | 554 | 53.01 |
| Primary school/no schooling | 64 | 6.12 |
| I don't know | 125 | 11.96 |
| Maternal caregiver occupation | | |
| Government officer/private sector officer | 298 | 28.52 |
| Self-employed | 101 | 9.67 |
| Primary caregiver/no paid work | 622 | 59.52 |
| Other (retired/don't know/other) | 24 | 2.30 |
| Main paternal figure | | |
| Birth father | 1,004 | 96.08 |

*(Continued)*

**Table 2.** (*Continued*)

| Variable | Frequency | Percentage % |
|---|---|---|
| Other family member (grandfather, stepfather, adoptive father, older brother, other family member)/other non-family member/none | 41 | 3.92 |
| Paternal caregiver education | | |
| University degree or higher | 169 | 16.17 |
| Diploma or certificate | 178 | 17.03 |
| Secondary school | 521 | 49.86 |
| Primary school/no schooling | 65 | 6.22 |
| I don't know | 112 | 10.72 |
| Paternal caregiver occupation | | |
| Government/private sector officer | 598 | 57.22 |
| Self-employed | 352 | 33.68 |
| Home maker/unemployed | 18 | 1.72 |
| Other (retired/don't know/other) | 77 | 7.37 |
| Parents are religious | | |
| Strongly disagree/disagree | 54 | 5.17 |
| I don't know | 82 | 7.85 |
| Strongly agree/agree | 909 | 86.99 |
| Number of siblings* | | |
| None | 104 | 10.18 |
| 1 | 520 | 50.88 |
| ≥2 | 398 | 38.94 |
| Sporting activities | | |
| Weekly | 199 | 19.04 |
| Monthly | 76 | 7.27 |
| Rarely | 536 | 51.29 |
| I don't engage in sporting activities | 234 | 22.39 |
| Non-physical extracurricular activities | | |
| Weekly | 412 | 39.43 |
| Monthly | 89 | 8.52 |
| Rarely | 433 | 41.44 |
| I don't engage in aesthetic activities | 111 | 10.62 |
| Alcohol consumption | | |
| Yes | 26 | 2.49 |
| No/prefer not to say | 1,019 | 97.51 |
| Smoking | | |
| Yes | 8 | 0.77 |
| No/prefer not to say | 1037 | 99.23 |
| Social media use | | |
| Daily | 623 | 59.62 |
| Weekly | 86 | 8.23 |
| Monthly/rarely/I don't use social media | 336 | 32.15 |

*(Continued)*

**Table 2.** (Continued)

| Variable | Frequency | Percentage % |
|---|---|---|
| Family happiness* | | |
| Very happy/happy | 995 | 97.45 |
| Unhappy/very unhappy | 26 | 2.55 |
| Caregiver alcohol consumption* | | |
| Yes | 39 | 3.82 |
| No/I don't know | 983 | 96.18 |
| Caregiver drug use* | | |
| Yes | 31 | 3.03 |
| No/I don't know | 991 | 96.97 |

*Missing: Siblings = 23; Family happiness = 24; Caregiver alcohol consumption = 23; Caregiver drug use = 23.

**Table 3.** Symptoms of depression and psychological distress experienced by Sri Lankan adolescents

| | PHQ-9 | K10 |
|---|---|---|
| Number of participants | 1,045 | 1,044 |
| Mean score | 6.61 | 18.10 |
| Standard deviation | 4.57 | 6.55 |
| Min | 0 | 0 |
| Max | 25 | 40 |
| % Above cut-off | (≥10) 24.11 | (≥18) 60.10 |
| % Severity | Non-minimal symptoms (≤4): 36.84 Mild symptoms (5–9): 39.04 Moderate symptoms (10–14): 18.37 Moderately severe symptoms (15–19): 4.11 Severe symptoms (20–27): 1.63 | Low distress (0–5): 5.75 Moderate distress (6–11): 9.20 High distress (12–19): 38.7 Very high distress (20–40): 46.36 |

administered the survey to ensure accuracy and comprehensiveness in translation and administration.

Additionally, adolescent mental health is a growing area of research, particularly in LMICs. This study provides timely research that aligns with the findings of the Lancet Commission on Adolescent Health and Wellbeing.

Nevertheless, we acknowledge some limitations. First, some tools used to measure prevalence and identify determinants associated with MHPs in the population have not been formally validated against a local gold-standard diagnostic interview. We relied on cut-off scores established elsewhere, which means that prevalence may have been under- or overestimated, and in general, self-reported symptom checklists of MHPs yield higher estimates of problems. Second, most participants were Sinhalese (99%) and Buddhist (96%). Further, some studies indicate high school dropout rates (Nanayakkara, 2020; Mayadunne and Kariyasekara, 2021) predominantly among adolescents from low socio-economic backgrounds (Nanayakkara, 2020), with poverty being the reason (Mayadunne and Kariyasekara, 2021). It is possible that findings of this study do not represent the experiences of all groups of adolescents in the multi-ethnic, multi-religious, socio-economically diverse Sri Lankan populations. Third, this study was conducted in one district only. The

results may not be able to be generalisable to the wider Sri Lankan adolescent population. Adding to this, we were not able to include participants from the north and north-eastern provinces of Sri Lanka which were most severely affected by armed conflict and instability. These populations may experience different MHPs and socio-economic factors, which may influence their mental well-being. Fourth, all young people in Sri Lanka have experienced geopolitical instability and conflict, so we could not make comparisons between a group that had or had not experienced these factors. Finally, this study only examined a small set of MHPs and determinants, and it is possible that relevant conditions, including those related to trauma, were overlooked.

Overall, however, we believe that the strengths outweigh these limitations and that the findings provide Sri Lankan policymakers with a useful indication of the burden and potentially modifiable risk and protective factors for MHPs experienced by adolescents in the country.

## Prevalence

In this study, 24.11% of Sri Lankan adolescents reported high levels of clinically significant depressive symptoms, and 60.10% reported extremely high levels of clinically significant symptoms of psychological distress. These findings indicate that many Sri Lankan adolescents are experiencing sub-optimal functioning through having low mood, low energy, being worried or feeling limited hope or optimism. For school-going adolescents, these factors have implications for concentration and learning capacity. Adolescent mental health and academic outcomes are interlinked, with each influencing or being influenced by the other (Lee et al., 2024). Children and adolescents with MHPs often have poorer academic outcomes (Lee et al., 2024), highlighting that long after, mental health is not only important for health but also for the education of Sri Lankan adolescents.

Comparing the findings to international contexts, our findings are inconsistent with previous international studies examining depressive symptoms in adolescents using the same tool (PHQ-9) (Kroenke et al., 2001). Studies in high-income countries (HICs) described similar findings (Andreas and Brunborg, 2017). However, when compared to LMICs, the prevalence is lower in Sri Lankan adolescents. For example, school-going samples of Chinese and Nigerian adolescents exhibit prevalence of moderately severe and severe depressive symptoms at 5.2 and 5.1%, respectively, whereas Sri Lankan adolescents had slightly lower prevalence at 4.1 and 1.6%, respectively (Andreas and Brunborg, 2017). In regional populations of Indian adolescents, prevalence (13.3%) was found to be almost half that reported by Sri Lankan adolescents (24.11%) (Ganguly et al., 2013). Additionally, lower prevalence of mild depressive symptoms (15.8%) was identified among Sri Lankan adults in comparison to Sri Lankan adolescents (39.04%) (Hanwella et al., 2014). Inconsistencies may be attributable to the instrument not being adequately validated or culturally relevant to the study population, thereby providing an inaccurate estimate representation. Furthermore, the self-reported nature of the tool may have resulted in under- or over-reporting of the problem. Nevertheless, this tool provides an indication of distress among Sri Lankan adolescents.

When comparing this study's findings of psychological distress to similar HIC and LMIC populations, Sri Lankan adolescents reported extremely higher levels. For instance, studies on Australian adolescents using the K10 reported a lower prevalence (mean = 17.3) (Blake et al., 2023). However, aligning with the findings of the current study,

**Table 4.** Psychosocial determinants of depressive symptoms among Sri Lankan adolescents

| Variable | Regression coefficient | p-value | 95% Confidence interval Lower limit | Upper limit |
|---|---|---|---|---|
| Age (in years) | 0.08 | 0.496 | −0.15 | 0.30 |
| Sex | | | | |
| Male | Ref. | | | |
| Female | 1.25 | **<0.001** | 0.72 | 1.79 |
| Number of siblings | | | | |
| No siblings | Ref. | | | |
| 1 sibling | −0.42 | 0.330 | −1.26 | 0.42 |
| ≥2 siblings | −0.08 | 0.855 | −0.95 | 0.78 |
| Engagement in sporting activities | | | | |
| Weekly | Ref | | | |
| Rarely/no engagement | 0.68 | **0.040** | 0.03 | 1.34 |
| Alcohol consumption | | | | |
| No | Ref. | | | |
| Yes | −0.35 | 0.706 | −2.14 | 1.45 |
| Smoking | | | | |
| No | Ref. | | | |
| Yes | 3.28 | **0.039** | 0.16 | 6.40 |
| Social media use | | | | |
| Rarely/Don't use | Ref. | | | |
| Daily use | 0.79 | **0.006** | 0.23 | 1.35 |
| Weekly use | 0.54 | 0.274 | −0.43 | 1.52 |
| Violent victimisation (JVQ) | | | | |
| Experiencing child maltreatment | 1.02 | **0.008** | 0.27 | 1.77 |
| Witnessing/Indirect victimisation | 0.78 | **0.005** | 0.23 | 1.33 |
| Experience of family violence | 0.98 | **0.005** | 0.30 | 1.67 |
| Are you currently living with: | | | | |
| Both birth parents | Ref. | | | |
| One/No birth parent or others | 1.16 | **0.009** | 0.29 | 2.03 |
| Maternal caregiver education level | | | | |
| University/Diploma | Ref. | | | |
| Secondary school | −0.57 | 0.146 | −1.34 | 0.20 |
| Primary/No schooling | 0.06 | 0.932 | −1.32 | 1.44 |
| I don't know | 0.11 | 0.860 | −1.13 | 1.36 |
| Maternal caregiver occupation | | | | |
| Government/Private sector | Ref. | | | |
| Self-employed | 0.10 | 0.845 | −0.90 | 1.10 |
| Homemaker/Unemployed | −0.37 | 0.295 | −1.07 | 0.32 |
| Retired/I don't know/other | −0.02 | 0.984 | −1.72 | 1.68 |

(*Continued*)

**Table 4.** (*Continued*)

| Variable | Regression coefficient | p-value | 95% Confidence interval Lower limit | Upper limit |
|---|---|---|---|---|
| Relationship with maternal figure (PBI) | | | | |
| Maternal care score | −0.15 | **<0.001** | −0.24 | −0.07 |
| Maternal overprotection score | −0.02 | 0.678 | −0.10 | 0.07 |
| Paternal caregiver education level | | | | |
| University/Diploma | Ref. | | | |
| Secondary school | 0.54 | 0.118 | −0.14 | 1.22 |
| Primary/no schooling | 0.84 | 0.209 | −0.47 | 2.15 |
| I don't know | −0.94 | 0.142 | −2.20 | 0.32 |
| Paternal caregiver occupation | | | | |
| Government/Private sector | Ref. | | | |
| Self employed | −0.01 | 0.972 | −0.58 | 0.56 |
| Homemaker/unemployed | 0.71 | 0.491 | −1.30 | 2.71 |
| Retired/I don't know/other | 0.60 | 0.244 | −0.41 | 1.61 |
| Relationship with paternal figure (PBI) | | | | |
| Paternal care score | 0.02 | 0.561 | −0.05 | 0.09 |
| Paternal overprotection score | 0.11 | **0.007** | 0.03 | 0.18 |
| Parents are religious | | | | |
| Disagree/Strongly disagree | Ref. | | | |
| I don't know | 0.49 | 0.493 | −0.91 | 1.88 |
| Agree/Strongly agree | 0.72 | 0.214 | −0.41 | 1.85 |
| Family is: | | | | |
| Happy/Very happy | Ref. | | | |
| Unhappy/Very unhappy | 1.53 | 0.076 | −0.16 | 3.22 |
| Academic stress (AESI) score | 0.23 | **<0.001** | 0.19 | 0.27 |
| Area of residence | | | | |
| Urban | Ref. | | | |
| Semi-urban | 0.52 | 0.114 | −0.12 | 1.17 |
| Rural | 1.22 | **0.001** | 0.47 | 1.97 |

*Note*: Numbers in **bold** indicate statistical significance. *AESI* Academic Stress Inventory; *JVQ* Juvenile Victimization Questionnaire; *PBI* parental bonding instrument; *Ref.* Reference Variable.

data from Indonesian adolescents from a similar LMIC socio-economic context to that of Sri Lankan adolescents reported a high prevalence of 31.7% (Tran et al., 2019). Variations in prevalence may be attributed to factors such as geographic location, which dictates the distribution of mental health resources, socio-economic circumstances of adolescents, exposure to trauma, as well as culturally derived illness beliefs (Kieling et al., 2011; Lund et al., 2018; Patel et al., 2018). Adolescents living in LMICs are disproportionately exposed to poverty, resource constraints and other socio-environmental stressors that likely contribute to higher prevalence and severity of psychological distress relative to their peers from HICs (Lund et al., 2018; Maddock et al., 2021; World Health Organization, 2023). In this study, the level of psychological distress experienced by Sri Lankan adolescents was higher than populations

**Table 5.** Psychosocial determinants of psychological distress among Sri Lankan adolescents

| Variable | Regression coefficient | *p*-value | 95% Confidence interval Lower limit | 95% Confidence interval Upper limit |
|---|---|---|---|---|
| Age (in years) | 0.53 | **0.004** | 0.17 | 0.89 |
| Sex | | | | |
| Male | Ref. | | | |
| Female | 0.98 | **0.026** | 0.12 | 1.85 |
| Number of siblings | | | | |
| No siblings | Ref. | | | |
| 1 sibling | 0.95 | 0.169 | −0.41 | 2.32 |
| ≥2 siblings | 1.85 | **0.010** | 0.45 | 3.25 |
| Engagement in sporting activities | | | | |
| Weekly | Ref. | | | |
| Rarely/no engagement | 1.78 | **0.001** | 0.72 | 2.83 |
| Alcohol consumption | | | | |
| No | Ref. | | | |
| Yes | 2.34 | 0.113 | −0.56 | 5.25 |
| Smoking | | | | |
| No | Ref. | | | |
| Yes | −1.15 | 0.653 | −6.20 | 3.89 |
| Social media use | | | | |
| Rarely/Don't use | Ref. | | | |
| Daily use | −0.25 | 0.594 | −1.15 | 0.66 |
| Weekly use | −0.19 | 0.817 | −1.76 | 1.39 |
| Violent victimisation (JVQ) | | | | |
| Experiencing child maltreatment | 1.37 | **0.027** | 0.16 | 2.58 |
| Witnessing/Indirect victimisation | 0.95 | **0.036** | 0.06 | 1.84 |
| Experience of family violence | 0.07 | 0.898 | −1.04 | 1.18 |
| Are you currently living with: | | | | |
| Both birth parents | Ref. | | | |
| One/No birth parent or others | 1.78 | **0.013** | 0.37 | 3.18 |
| Maternal caregiver education level | | | | |
| University/Diploma | Ref. | | | |
| Secondary school | 1.38 | **0.030** | 0.13 | 2.63 |
| Primary/no schooling | 0.58 | 0.611 | −1.65 | 2.80 |
| I don't know | 0.91 | 0.375 | −1.10 | 2.92 |
| Maternal caregiver occupation | | | | |
| Government/Private sector | Ref. | | | |
| Self-employed | −0.09 | 0.915 | −1.70 | 1.53 |
| Homemaker/unemployed | −0.36 | 0.531 | −1.48 | 0.77 |
| Retired/I don't know/other | −1.06 | 0.450 | −3.81 | 1.69 |

(*Continued*)

**Table 5.** (*Continued*)

| Variable | Regression coefficient | *p*-value | 95% Confidence interval Lower limit | 95% Confidence interval Upper limit |
|---|---|---|---|---|
| Relationship with maternal figure (PBI) | | | | |
| Maternal care score | 0.10 | 0.137 | −0.03 | 0.24 |
| Maternal overprotection score | 0.07 | 0.343 | −0.07 | 0.21 |
| Paternal caregiver education level | | | | |
| University/Diploma | Ref. | | | |
| Secondary school | 1.24 | **0.026** | 0.15 | 2.34 |
| Primary/No schooling | 2.55 | **0.018** | 0.43 | 4.68 |
| I don't know | −0.88 | 0.395 | −2.91 | 1.15 |
| Paternal caregiver occupation | | | | |
| Government/private sector | Ref. | | | |
| Self-employed | 0.53 | 0.258 | −0.39 | 1.44 |
| Homemaker/unemployed | 1.71 | 0.303 | −1.54 | 4.95 |
| Retired/I don't know/other | 0.76 | 0.360 | −0.87 | 2.40 |
| Relationship with paternal figure (PBI) | | | | |
| Paternal care score | 0.10 | 0.079 | −0.01 | 0.22 |
| Paternal overprotection score | 0.17 | **0.007** | 0.05 | 0.30 |
| Parents are religious | | | | |
| Disagree/Strongly disagree | Ref. | | | |
| I don't know | −1.03 | 0.370 | −3.28 | 1.22 |
| Agree/Strongly agree | −0.07 | 0.942 | −1.90 | 1.76 |
| Family is: | | | | |
| Happy/very happy | Ref. | | | |
| Unhappy/very unhappy | 1.91 | 0.170 | −0.82 | 4.65 |
| Increased academic stress (AESI) score | 0.10 | **0.004** | 0.03 | 0.17 |
| Area of residence | | | | |
| Urban | Ref. | | | |
| Semi-urban | 0.49 | 0.359 | −0.56 | 1.53 |
| Rural | 1.18 | 0.058 | −0.04 | 2.39 |

*Note*: Numbers in **bold** indicate significance. *AESI* Academic Stress Inventory; *JVQ* Juvenile Victimization Questionnaire; *PBI* Parental Bonding Instrument; *Ref.* Reference Variable.

from other LMICs. Our findings indicate that for Sri Lankan adolescents, these already existing determinants of MHPs faced by adolescents in LMICs are further exacerbated by additional systemic and social challenges posed by the remnants of a civil war and decades of socio-political instability. Young people in Sri Lanka bear an unfair mental health burden from ongoing crises facing the country and appear to be living in distress, with poor functioning and hopelessness for the future.

## Determinants

Several determinants acted as risk factors for developing MHPs. They can be categorised into the six groups discussed in the

conceptual framework of this study: individual factors, health risk factors, violent victimisation, family, school and community. Several of these can be classified as modifiable determinants, where psychosocial prevention interventions can be employed through universal, selective and targeted approaches (Lund, 2023). Other risk factors were less or non-modifiable and often caused by structural inequities, which require policy changes or innovations.

### Modifiable risk factors

*Health risk factors.* We identified several health risk factors, such as decreased physical activity, smoking and daily social media use. These risk factors are likely to be either bidirectional or MHPs preceded them, which can be seen as efforts to numb emotions or achieve social links.

Supporting this, many studies confirm the positive impact of physical activity on mental health. The reasoning is that physical activity releases neurotransmitters related to mood regulation, boosts self-esteem or acts as a buffer against stress (Rasalingam et al., 2022). However, no studies conducted on Sri Lankan adolescents examine the direct influence of smoking and social media use on adolescent mental well-being. The association between smoking and mental health in Sri Lankan adolescents is not causal, but rather a consequence of the inability to cope with frustrations and stress (Rodrigo, 2014). This is further highlighted by substance use being higher among young people living in war-affected areas. This is primarily due to the nature of uncertainty regarding their futures (Rodrigo, 2014).

The negative influence of social media on adolescents' body image is detailed in studies conducted on Sri Lankan adolescents (Lokumannage, 2020; Baminiwatta et al., 2021). Its impact on overall mental health needs further exploration.

*Violent victimisation.* This study specifically examined the impact of three forms of violent victimisation: experiencing child maltreatment, witnessing/indirect victimisation and exposure to family violence. All forms of victimisation were significantly associated with depressive symptoms. Child maltreatment and witnessing/indirect victimisation were significantly associated with psychological distress. In Sri Lanka, there have been several large-scale studies spanning decades that highlight child abuse being a public health problem in the country (Chandraratne et al., 2018). These data indicate a high prevalence of violence against children, the lifetime history of experiencing violence and the continuous nature of violence (Sriskandarajah et al., 2015b; Chandraratne et al., 2018). Moreover, prolonged exposure to violence perpetrated against young people or witnessing violence at a young age affects the developing brain (Fernando and Karunasekera, 2009). For example, this can lead to social, emotional or cognitive problems later in life or to behaviours that cause disease and injury (Fernando and Karunasekera, 2009). Among school-going children in Sri Lanka, specific acts of violence, such as corporal punishment, has shown to directly predict the extent a child would be psychologically maladjusted (De Zoysa et al., 2006; De Silva, 2007; Chandraratne et al., 2018). Non-parent-to-child violence (i.e., school, peer and community) has also been found to affect Sri Lankan children psychologically (De Zoysa et al., 2006; De Silva, 2007; Chandraratne et al., 2018).

A study conducted in 2006 on Sri Lankan university students (Fernando and Karunasekera, 2009), using the same tool used in this study (JVQ), identified child maltreatment being significantly higher among males and witnessing violence at home being the highest form of indirect victimisation (Fernando and Karunasekera, 2009). Supporting these findings, in this study, overprotective paternal figures were significantly associated with higher MHPs. Here, overprotectiveness refers to authoritarian or controlling behaviour (Parker et al., 1979; Kendler, 1996). Many Sri Lankan children appear to be victims of violence in environments often supposed to protect them (Fernando and Karunasekera, 2009).

A more recent study on child abuse in Sri Lanka, conducted in 2018 on the same population of late-adolescents from Gampaha District, highlights a high prevalence of physical, emotional and sexual abuse (Chandraratne et al., 2018). At a regional level, data on violent victimisation mostly appear in Indian studies. Again, revealing a high prevalence of physical and sexual abuse, as well as neglect (Chandraratne et al., 2018). Comparisons between the data from this study and those of Indian studies may be precluded due to varying definitions of 'violent victimisation' and 'child abuse' and the tools used for data collection. Of noteworthy importance, evidence suggests that children may deny or hide incidents of violent victimisation, particularly in Asian countries, due to shame; therefore, this study may have underestimated experiences of violent victimisation (De Zoysa et al., 2006).

*Family environment.* Several factors in the family environment of a Sri Lankan adolescent are significantly associated with mental health outcomes: adolescents who did not live with both birth parents reported higher levels of MHPs, and paternal and maternal education levels below secondary education were associated with higher psychological distress.

The role of family structure and parental education levels has been extensively researched globally as playing a role in mental health outcomes of adolescents (Langton and Berger, 2013; Behere et al., 2017; Xiang et al., 2024). This research indicates that adolescents in two-biological-parent families or with parents who have higher education levels have better mental health outcomes than their counterparts (Langton and Berger, 2013; Behere et al., 2017; Xiang et al., 2024). Adolescents living in two-biological-parent families are considered to have greater economic resources, closer emotional ties to parents and less experience of stressful events (Langton and Berger, 2013; Behere et al., 2017). Similarly, higher education levels are often considered indicators of socio-economic status, thereby parents with higher education are considered as more likely to create a favourable home environment, especially in terms of economic conditions, resources and opportunities (Xiang et al., 2024).

In South Asian societies, the principal source of emotional, financial and social support is provided by family (South Asian Health Hub, 2024). Multigenerational households, close-knit families and shared responsibilities are common (South Asian Health Hub, 2024). Supporting this, a recent Sri Lankan study on school-going adolescents identified that not having a close family member to discuss problems with was significantly associated with emotional and behavioural problems (Nadeeka and Wijewardena, 2023). These data indicate that the combined effects of a lack of economic and emotional support created by changing family structures and limited educational opportunities for caregivers are contributing to poorer mental health outcomes in Sri Lankan adolescents. Notably, in resource-constrained settings, family dynamics are greatly influenced by social and systemic challenges that are often beyond individual control. For example, caregivers' lack of education is likely to be a direct result of limited educational opportunities available to older generations at a time when Sri Lanka was enduring conflict.

In this study, having a caring maternal figure positively correlated with lower depressive symptoms. Here, maternal care is characterised as warmth, closeness, trustworthiness and affection

(Parker et al., 1979; Kendler, 1996). This finding is consistent with prior research conducted on Sri Lankan children subject to trauma, using the same instrument (PBI) (Sriskandarajah et al., 2015a). It highlights positive parenting behaviours, such as maternal care here, as a powerful protective factor that mitigates the adverse effects of trauma on children's mental health (Sriskandarajah et al., 2015a). Considering that violent victimisation is strongly correlated with poor mental health outcomes among adolescents in this study, the positive elements of this protective factor highlight an opportunity to enhance existing effective parenting practices that may be relevant to building mental health resilience and promoting positive mental health to adolescents.

*School environment.* In this study, increased academic stress was associated with significantly more severe MHPs, and increased age was associated with higher psychological distress. Sri Lanka has compulsory education until the age of 16 years, and adolescents are exposed to two national-level examinations (Abayasekara and Arunatilake, 2018). These findings may be attributable to the increased school-related workload and stress (Rodrigo et al., 2010).

Education is often seen as a pathway out of poverty, particularly in LMICs, where education can be used as a lever against overcoming systemic barriers that may hinder the development of individuals, households, communities and societies (United Nations Educational, 2017; Nanayakkara, 2020). As a result, Sri Lankan adolescents carry generational expectations of educational attainment and parental pressure to do well in schooling. Adding to this, parents who may not have previously had higher levels of education may also have unrealistic expectations of schooling systems, thereby placing additional pressure on adolescents to excel at all aspects of education. Subsequently, this can contribute to adolescents feeling shame in their academic achievements and in reaching out for help, further exacerbating MHPs. This is supported by the finding that lower levels of parental education were associated with higher psychological distress. The current socio-economic challenges in Sri Lanka add extra burdens to adolescents already facing extreme expectations and pressures.

### Less-modifiable risk factors

*Individual factors.* Being female was strongly associated with higher levels of MHPs, supporting evidence from LMICs that girls are more likely to experience MHPs than boys (Rasalingam et al., 2022; Shah et al., 2024). The onset of MHPs in adolescent girls is multifactorial (Michelini et al., 2021). For Sri Lankan adolescent girls, cultural and gender norms may shape mental health outcomes (Chandradasa and Rathnayake, 2019; Rasalingam et al., 2022). Programmes and interventions addressing such entrenched norms may benefit the mental health of adolescent girls (Shah et al., 2024).

Sri Lankan adolescents who had two or more siblings had higher psychological distress. Adolescence is a period where opinions of peers start to take precedence over those of parents (Patel et al., 2018), and siblings become important influencers (Pathirana, 2016). Of six major relationships (mothers, fathers, siblings, grandparents, friends and teachers), early adolescents perceived conflict occurring most frequently with siblings (Pathirana, 2016). This was predominantly attributed to seeking autonomy (Pathirana, 2016). Studies conducted on Sri Lankan adolescents examining sibling relationships highlight similar findings of conflicts with siblings, primarily due to personal space and sharing (Pathirana, 2016). Adding to this, the lack of personal bedrooms and crowded household composition worsened by economic disadvantage is likely to exacerbate conflicts with siblings and consequently MHPs.

*Community.* At a community level, rural living was significantly associated with higher levels of depressive symptoms in Sri Lankan adolescents. This finding is somewhat inconsistent with international evidence that often associates urban living, comprised of concentrated poverty, low social capital and social segregation, with a higher risk of poor mental health outcomes (Hannon et al., 2024). The opposite finding in this study may be a wider indication of poverty as a proxy indicator of MHPs in Sri Lankan adolescents. Compounding this, Sri Lanka is characterised by numerous barriers in supply and access to mental healthcare (Chandradasa and Kuruppuarachchi, 2017). Care is concentrated in urban areas, leaving the rural population with limited access (Chandradasa and Kuruppuarachchi, 2017). In addition, the shortage of resources available to help alleviate the mental health burden is further magnified by the lack of experts, where only 0.29 psychiatrists are available per 100,000 people (Chandradasa and Kuruppuarachchi, 2017). It is also evident that in Sri Lanka, the mental health needs of adolescents are not considered separately from those of adults (Rajapakshe et al., 2023), as is the case in many resource-constrained settings (Zhou et al., 2020).

### Implications and conclusion

Overall, we identified a high prevalence of depressive and psychological distress symptoms experienced by adolescents living in Sri Lanka. These MHPs were multifactorially determined and were identified in multiple environments directly and indirectly influencing Sri Lankan adolescents. Greater focus on addressing determinants of MHPs will enable research, policy and practice to be developed with inter-sectoral and inter-disciplinary collaboration, where different arms of government and civil society work to address not just the consequences of MHPs, but the underlying determinants of them. It was noteworthy throughout this study that Sri Lankan adolescents face additional social and systemic challenges as a result of continuous crises affecting the country as a whole. Stability and security in the country are essential to improving mental health outcomes in this underserved population.

Modifiable risk factors provide a potential starting point to develop mental health policies, programmes and research that can build mental health resilience and provide adolescents with the necessary tools to navigate their mental well-being. Less-modifiable factors provide a potential point for national-level policy change. Given that the current health infrastructure in the country is very resource-poor, focusing on primary prevention, including programmes based in communities and schools, may be key.

1. At an individual level, gender is a less-modifiable risk factor. Addressing this will require robust, enforceable national policies, strategies and frameworks that specifically target the mental health of girls. These can then be operationalised through community or school-level life-skill programmes and other interventions that align with the national framework and work to improve the mental health of adolescent girls. Evidence does suggest that, when operationalised, interventions including both adolescent boys and girls are more effective than interventions that target girls alone (Shah et al., 2024).
2. Several modifiable health risk factors can be addressed through public health programmes and research. School-based physical education programmes may be influential in addressing the effects of limited physical activity on the mental health of Sri

Lankan adolescents. However, here, resource limitations in rural schools will need to be taken into consideration. Further, public health research on social media use and its association with mental health that is contextualised to the Sri Lankan adolescent population can provide valuable insight.

3. Violent victimisation requires further public health research using uniform definitions and tools to understand its true burden (Chandraratne et al., 2018).

4. In the family environment of Sri Lankan adolescents, caregiver education programmes administered through schools or the community by practitioners and researchers can provide caregivers with skills and knowledge to practice warmth, closeness, trust and affection distinguished from authoritarian and controlling behaviours. In addition, these programmes can be an avenue for researchers to identify already existing, effective positive parenting behaviours and their effects on adolescent mental well-being. Thereby, already existing practices can be further strengthened and amplified. Such programmes can also provide an opportunity for practitioners and advocates to educate caregivers about violence against young people and its impact on mental health.

5. The school system, as it stands, has a strong focus on academic performance. A growing body of research endorses the need for a population approach to mental health promotion, which can perhaps be employed directly to Sri Lankan adolescents by the education system through school-based mental health programmes embedded in the curricula (Harte and Barry, 2024). Given the mental health implications of a highly stressful academic environment, such programmes can provide Sri Lankan adolescents with adequate support for school-related stress and distress, enabling them to manage their mental well-being while simultaneously excelling in academic life.

6. At a community level, the area of residence is less-modifiable. National-level mental health policies that address accessibility, affordability and availability of mental healthcare specifically to rural communities are essential. Furthermore, the mental health of adolescents needs to be addressed separately from that of adults.

These primary prevention methods further highlight the need for key stakeholders, such as educators, caregivers, policymakers and researchers, to recognise risk and protective factors of mental health in an adolescent's immediate and surrounding environments, and to work towards implementing supportive strategies within their respective roles.

Findings from this study will be shared in a report format with the schools involved in this project, the educational departments involved in the facilitation of this project and the Ministry of Education, Sri Lanka. Additionally, findings will be shared in publications in peer-reviewed journals.

**Open peer review.** To view the open peer review materials for this article, please visit http://doi.org/10.1017/gmh.2025.10055.

**Supplementary material.** The supplementary material for this article can be found at http://doi.org/10.1017/gmh.2025.10055.

**Data availability statement.** The datasets used and/or analysed during the current study are available from the corresponding author on reasonable request.

**Acknowledgements.** We thank the secondary schools involved in this study and their staff and students. We extend thanks to the Ministry of Education, Sri Lanka, and the Educational Directors of Gampaha District who facilitated the success of this study.

**Author contribution.** C.M. Contributed to conceptualisation, data collection, data analysis, data interpretation, writing the original draft and reviewing and editing manuscripts. M.C. Contributed to conceptualisation, data collection, data interpretation, reviewing and editing manuscripts and supervision. T.T. And J.F. Contributed to conceptualisation, data analysis, data interpretation, reviewing and editing manuscripts and supervision. J.A. Contributed to conceptualisation, data interpretation, reviewing and editing manuscripts and supervision. S.S. Contributed to data collection, data interpretation and reviewing manuscripts.

**Financial support.** C.M. is supported by a Monash University Research Training Program Scholarship and a Monash University Travel Grant. JA received an NHMRC EL1 Fellowship. J.F. is supported by the Finkel Professorial Fellowship, which is funded by the Finkel Family Foundation. The funders had no role in the study design, data collection, data analysis, data interpretation or writing.

**Competing interests.** The authors declare none.

**Ethics statement.** First, ethics approval was obtained from the University of Kelaniya, Faculty of Medicine Ethics Review Committee (ID: P/124/09/2022). Next, the project was registered with the Monash University Human Research Ethics Committee (ID: 37225). Third, approval to conduct fieldwork was obtained from the Department of Education, Western Province, Sri Lanka (ID: WP/ED/DEV/13/V); the Line Ministry of Education (ID: ED/03/56/02) and the Zonal Education Office of Gampaha Zone, Gampaha District, Sri Lanka.

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
