## [Reviewer Report]

This paper addresses a timely and necessary topic in both global and local contexts, with a clearly identified research gap.

Minor revisions are recommended:

1. Did the study consider incorporating Patient & Public Involvement (PPI) measures? Additionally, can the paper discuss how findings will be shared with participants, schools, parents, education systems, and mental health practitioners in accessible, non-stigmatizing formats? Emphasizing actionable steps to reduce harm and improve adolescent mental health, rather than solely presenting clinical findings, would strengthen the impact of this study.

2. Were any paper-based responses collected by teachers? If so, what measures ensured data confidentiality and privacy, given that school systems in Sri Lanka have been noted for breaching such boundaries?

3. Can the discussion section further explore the protective factors identified in the study? Emphasizing these positive elements can reinforce existing effective practices and knowledge while providing clearer guidance for practitioners, researchers, and advocates working to enhance adolescent mental health support.

4. Consider adding a concluding sentence after the list of primary prevention methods that serves as a call to action, encouraging caregivers, educators, policymakers, and advocates to recognize risk factors and implement supportive measures within their respective roles.

5. Suggested keyword revision: Use “adolescent mental health” instead of “child mental health.”

6. Minor typographical errors are present and should be corrected.

---

## [Reviewer Report]

The study assessed mental health problems in Sri Lankan adolescents and the associated factors. The authors acknowledge that prior research in the area had been done and this was an update of the literature, hence not novel.

Some comments for consideration;

1. In the abstract methods, state what tests were used and what analyses were done

2. In the abstract results, start with descriptive statistics of the sample

3. In the main text, why did they not do a national survey, why this singular district?

4. Finally, what criteria was used to determine which variables enter the multivariate model? This needs to be revised and will most likely affect the results and discussion.

---

## [Editor Report]

Dear Chethana Mudunna,

Your Manuscript ‘Prevalence and Determinants of Mental Health Problems Experienced by School-Going Adolescents in Sri Lanka’ has now been reviewed,

---

## [Reviewer Report]

The authors have addressed all previous comments. I appreciate the authors' careful attention to the reviewer feedback. The revised manuscript is strengthened by highlighting a clear call to action for stakeholders based on the findings. I recommend acceptance.

---

## [Editor Report]

Dear Mudunna, Chethana ,

Your revised manuscript ‘Prevalence and Determinants of Mental Health Problems Experienced by School-Going Adolescents in Sri Lanka’ has now been reviewed,